# Maximum trunk tip force assessment related to trunk position and prehensile 'fingers' implication in African savannah elephants

**Pauline Costes**[1,2]*, **Arnaud Delapré**[2], **Céline Houssin**[2], **Baptiste Mulot**[3], **Emmanuelle Pouydebat**[1☯], **Raphaël Cornette**[2☯]

**1** Adaptive Mechanisms and Evolution, UMR 7179 CNRS/MNHN, Paris, France, **2** Institut de Systématique, Evolution, Biodiversité, UMR 7205 CNRS/MNHN /SU/EPHE/UA, Paris, France, **3** ZooParc de Beauval & Beauval Nature, Saint-Aignan, France

☯ These authors contributed equally to this work.
* pauline.costes0@gmail.com

**Data Availability Statement:** All relevant data are available in the public repository Open Science Framework (DOI: 10.17605/OSF.IO/85KJ3).

## Abstract

African elephants have a wide range of abilities using their trunk. As a muscular hydrostat, and thanks to the two finger-like processes at its tip, this proboscis can both precisely grasp and exert considerable force by wrapping. Yet few studies have attempted to quantify its distal grasping force. Thus, using a device equipped with force sensors and an automatic reward system, the trunk tip pinch force has been quantified in five captive female African savanna elephants. Results showed that the maximum pinch force of the trunk was 86.4 N, which may suggest that this part of the trunk is mainly dedicated to precision grasping. We also highlighted for the first time a difference in force between the two fingers of the trunk, with the dorsal finger predominantly stronger than the ventral finger. Finally, we showed that the position of the trunk, particularly the torsion, influences its force and distribution between the two trunk fingers. All these results are discussed in the light of the trunk's anatomy, and open up new avenues for evolutionary reflection and soft robot grippers.

## Introduction

African savannah elephants (*Loxodonta africana*) use their trunk in various contexts such as feeding, watering, environmental exploration, vocalisation, social behaviour, tool making and tool use [1]. This multitasking proboscis is mainly composed of six muscle groups: longitudinal, superficial oblique, deep oblique, radial dorsal, radial ventral; and transversal muscle groups [2]. It has 63,000 facial neurons [3] and has no skeletal support system. Asian elephants (*Elephas maximus*) possess up to 150,000 muscle fascicles but this data has not been reported for African savannah elephants [4]. This particular anatomy makes this organ a muscular hydrostat, i.e., an organ that retains its volume regardless of shape change. Its high degree of freedom [5] allows it to move precisely and exert significant force [6–8]. African elephants have colossal strength, capable of uprooting a tree, yet few studies have attempted to quantify this grasping force. Experimentally, it has been shown that a female Savannah African elephant can lift 63 kg by wrapping her trunk around a dumbbell, which is only 2% of her body weight

**Funding:** This work was supported by grants IBEES 20SB403U7209, Association Sorbonne Université (EP) and CNRS 80 PRIME, Centre National de la Recherche Scientifique (EP), which we thank. The funders had no role in study design, data collection and analysis, decision to publish, or preparation of the manuscript.

**Competing interests:** The authors have declared that no competing interests exist.

and 65% of her trunk weight. However, the authors assume that the elephant is limited by the experiment's setup and can lift a heavier weight [7]. The trunk tip of the African elephants, unlike the Asian elephants, has two finger-like processes on the dorsal and ventral trunk edges [9]. The dorsal tip finger is a continuation of the dorsal part of the trunk, with radial and longitudinal muscles extending almost to the very distal end. On the ventral side, superficial oblique muscles become more longitudinally aligned close to the tip, whereas deep oblique muscle fibres seem to gradually align and merge with the radial ventral muscles [2]. Thus, they can grasp objects of various shapes, sizes, weights and quantities with their trunks by adapting their grasping strategy [2, 7–12]. Indeed, they can grasp small or thin objects using suction [11] and use their trunk fingers to pack together a series of items for a single pickup varying their force from 7 to 47 N [8]. African elephants tend to pinch smaller objects between the two trunk-tip fingers and grasp larger objects by wrapping the trunk distally around them [2]. During the test sequence to validate our experimental system, it was shown that a female Savannah African elephant can apply a force of at least 32.79 N when pinching with the two fingers of its trunk [6].

The grasping systems of animals, including elephants, have been selected through evolution for their effectiveness in responding to functional, morphological and environmental constraints. These capabilities have inspired robotic engineers to incorporate bioinspired technologies into their designs [13–16]. Thus, for the past two decades, the full range of abilities of the elephant's proboscis has inspired research in soft robotics, specifically in manipulative and grasping robots. It is the field of robotics in which the actuators and electronics are engineered from elastomers, textiles and other soft materials to mimic the flexibility and deformability of natural biological tissue, [17–21]. Actually, grasping objects is one of the fundamental capabilities that robots must be able to perform to carry out routine activities. Grippers are thus the most essential components of robots. They are essential in many manipulation tasks, serving as end-of-arm tools and mechanical interfaces between robots and environments/grasped objects [22]. Biomimetic soft elephant grippers are developed to enhance human safety and adapt to cluttered and unpredictable environments, by actively or passively reconfiguring their shape, for adaptation of their elastic bodies to the objects they interact with [23, 24].

In a biomimetics context, the force that elephants can exert is important data to collect. It is important for robotics and also for evolutionary biology in the context of morpho-functional studies linking performance, e.g. bite force [25] or grip force [26], and the type of grasping or morphology of the organ. Therefore, this study focuses on the pinching grasp exerted by the prehensile trunk tip because this allows small objects to be picked up with high precision [2, 8, 11], which is complex and particularly useful in robotics.

Using a device equipped with force sensors and an automatic reward system [6], the maximum trunk tip pinch force has been quantified in five captive female African savanna elephants (ZooParc of Beauval, France). As previously mentioned, it was demonstrated that an elephant can apply a pinching force of at least 32.79 N [6]. Thanks to the keepers' indications, we know that the force of the tip of the trunk is much less than that generated by a human hand grip which ranges between 274 N for women and 461 N for males. In comparison, human tip pinch, i.e. thumb tip to index fingertip, force is between 49 N for women and 68 N for males [27]. Thus, we hypothesise that the maximum trunk tip force ranges between 32.79 N and 257 N.

We studied the force of pinches performed in different trunk positions using several sensors' orientations. In humans, arm position, such as wrist flexion and forearm rotation, affects the generation of grip force [28, 29]. Thus, despite a radically different structure, the position of the elephant's muscular hydrostat could impact the deployment of the force. The distribution of the applied force between the two trunk fingers was also studied as the system has two

**Table 1. Age and origin of the studied elephants.**

| Name | Birth location | Birth year | Age (in 2023) |
|------|---------------|-----------|---------------|
| Marjorie | Born wild at Kruger National Park | 1986 | 37 yrs |
| M'Kali | Born wild at Kruger National Park | 1989 | 34 yrs |
| Tana | Born wild at Unknown | 1987 | 36 yrs |
| Juba | Born wild at Unknown | 1987 | 36 yrs |
| Ashanti | Born captive-born by Kruger x Shaba at Knowsley Safari Park | 2003-01-15 | 20 yrs |

independent sensors. The dorsal finger, pointy and longer, has a relative volume of the trunk muscles bigger than the ventral finger, rounded and shorter [2]. Therefore, we expect the dorsal finger to exert more force than the ventral finger.

All results will be discussed from an evolutionary and bioinspired perspective.

## Materials and methods

### Subjects and housing

The experiments were conducted at the ZooParc de Beauval (France) for 3 months, in May and July 2022 and then in April 2023. Information concerning the involved female African savannah elephants is available in Table 1.

In this study, we observed elephants in an indoor setting. Every morning, time was devoted to medical training: one to three elephants were isolated in an enclosure dedicated to daily care and grooming, to facilitate any future medical intervention. The experiment took place during those training sessions and the running order changed each week. Before the experiment, the elephants had already been familiarised by the keepers with fake pinch sensors (S5 Fig).

### Ethical note

As the data were collected by keepers during the daily elephant training sessions, only standard elephant-keeper interactions occurred, following zoo security regulations. All procedures were conducted following the relevant CNRS guidelines and European Union regulations (Directive 2010/63/EU).

### Functioning of the device

The device for measuring the grasping force was a wooden box 1.60 m high, with two force sensors connected to an electronic system (Fig 1). The latter recorded the pinching force and automatically released apples as a reward. Force thresholds were defined from 0.5 to 8 kg. Depending on the ability of the elephants to pinch on the sensors, thresholds were every 500g or every kilo. When the individual achieved to pinch hard enough to pass a threshold an apple was released. To release the next apple, she had to pinch harder than the previous time and exceed a new threshold. By repeating this action, maximum force was achieved. Details of the design and functioning of this device are set out in the article dealing with the validation of this system in a zoological park [6]. Unlike the system presented in the latter article, the sensors were upright at the height of the trunk when extended right in front of the elephant, i.e. 1.20 m high (Fig 2A).

During the first three weeks of May 2022, the elephants were trained to experiment without being filmed. Those data were not used for analyses. Then, during one week in May and the whole month of July 2022, the elephants experimented with the sensors in a vertical orientation. Then, to test the influence of the position of the trunk on the maximum force and the

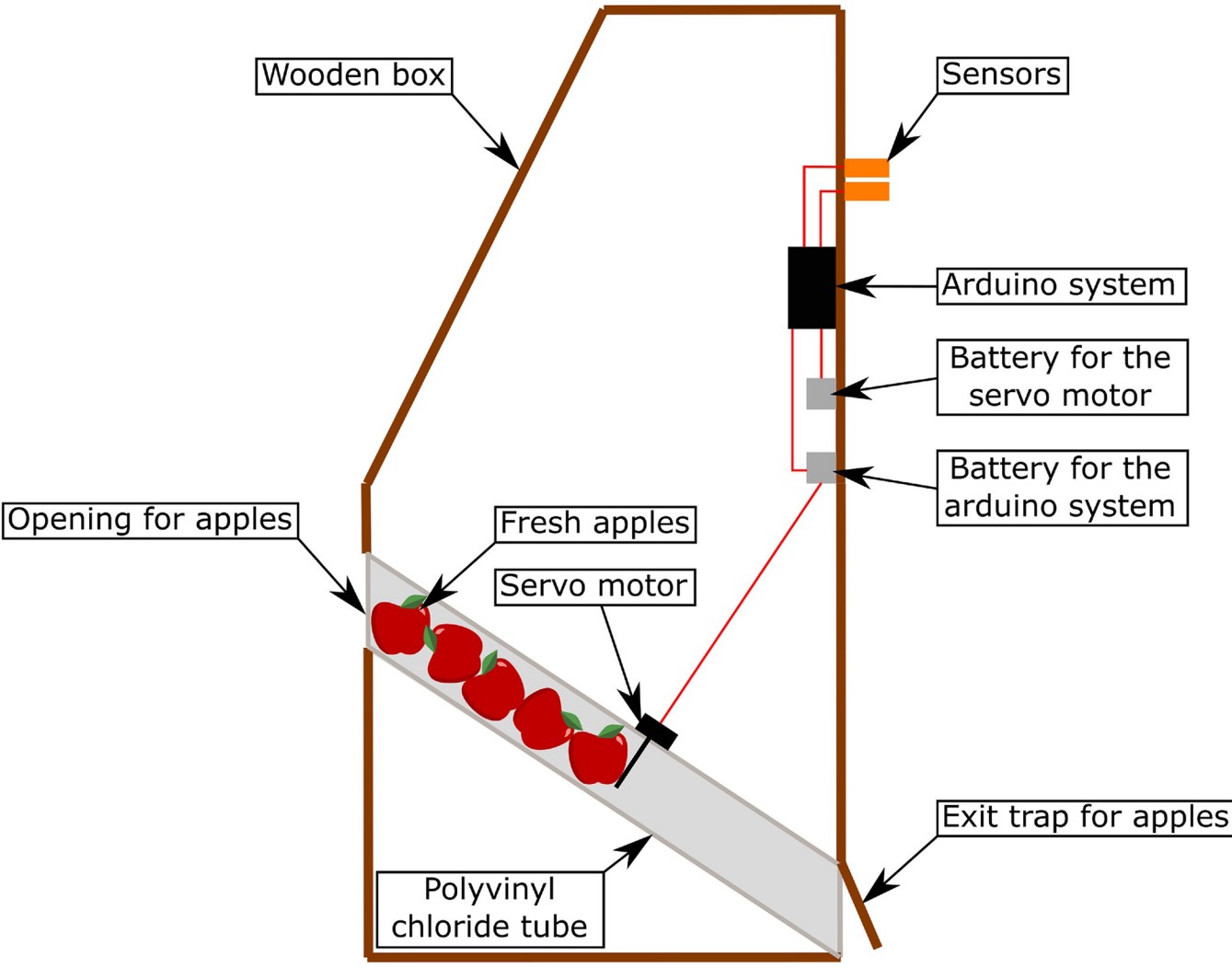

**Fig 1. Schema of the experimental device seen from the side.** See Cornette et al. 2022 [6] for more details.

distribution of force between the two fingers, the sensors' orientation was changed to horizontal and the elephants experimented with it for the whole month of April 2023 (Fig 2B).

## The course of the experiment

We conducted three to nine experimental sessions per elephant (see details in Table 2). An experimental session was conducted as follows: each sensor is calibrated every day, before the experiment, using three objects with known masses [for more details on the calibration protocol, see 6]. Then, the elephant was isolated in a training pen and kept away from the box by a keeper. The sensors have been placed at a distance of 2 ± 0.1 m from the fence. The computer program has been started, the code used to measure the forces has been sent to the Arduino system, the fresh apples have been loaded into the box and the camera that filmed the experiment from a side view has been started. The signal has been given to the keeper who suggested that the elephant come and pinch the sensors, with the spoken command: "pinch". The elephant pinched the sensors harder and harder and was rewarded with apples. The force data has been recorded live on the computer. When the elephant was no longer able to reach the

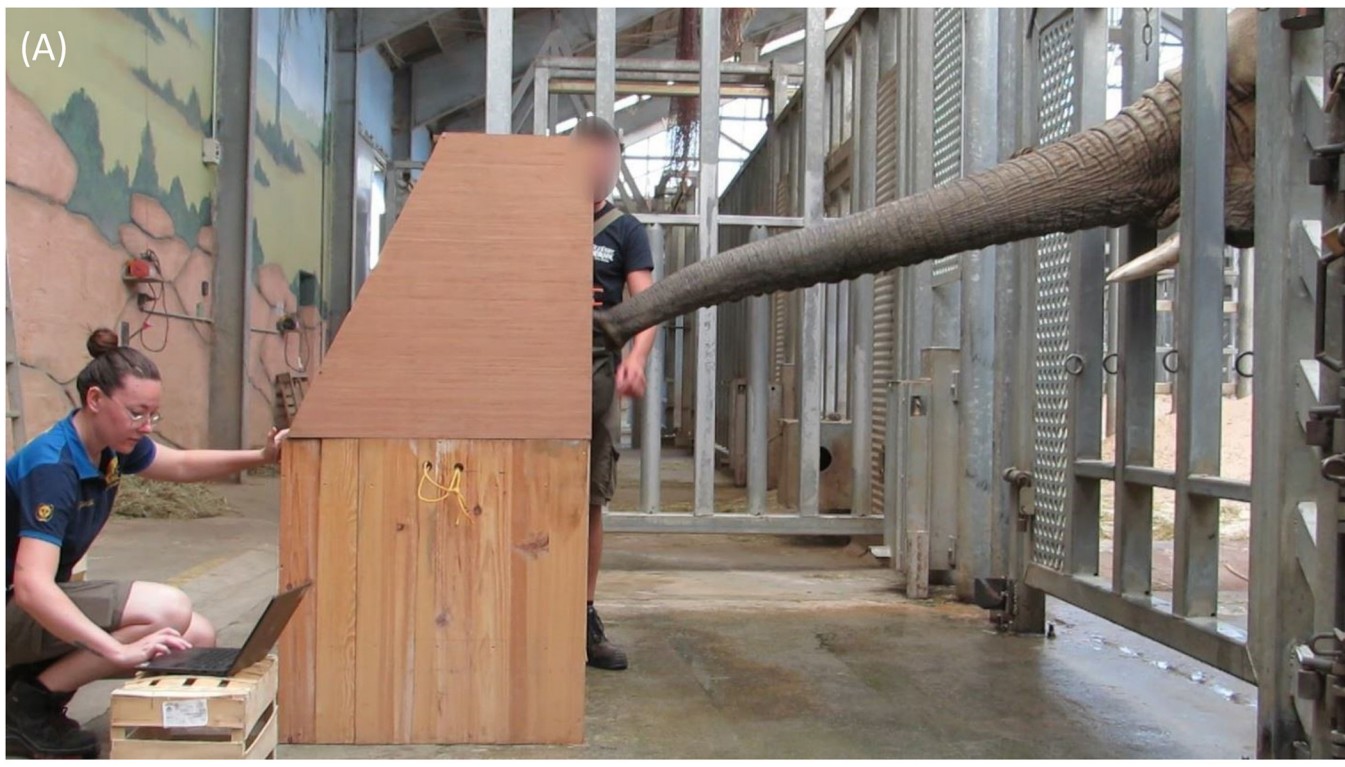

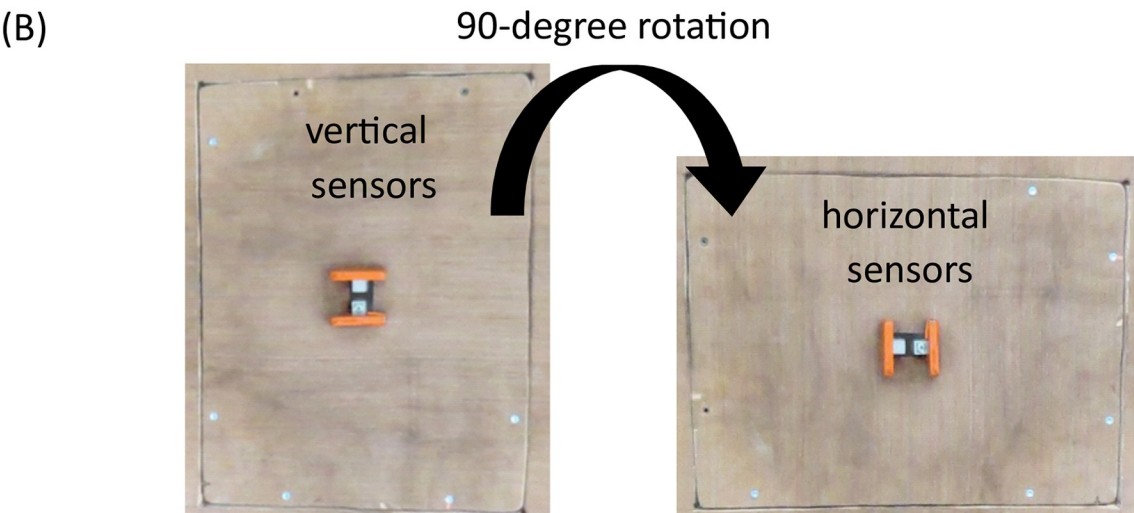

**Fig 2. Photos of the experimental device.** Side view of the whole setup when the elephant searches for the vertical sensors with its trunk straight (A). Change of sensors' orientation view from the front: from vertical to horizontal orientation (B).

next threshold and therefore began to lose interest or become annoyed by pushing, tapping or pulling on the device, the force thresholds were reset. Thus, the elephant could get an apple easily and end up with a good experience giving her the desire to repeat the experiment in the next session. Then the session stopped. The keeper removed the elephant, the device was removed and the camera was turned off. The sessions lasted between 1 min 35 sec and 7 min

**Table 2. The number of sessions by elephants according to the orientation of the sensors.**

|  | Vertical sensors | Horizontal sensors |
|---|---|---|
| Tana | 6 | 7 |
| M'Kali | 5 | 9 |
| Ashanti | 5 | 7 |
| Juba | 3 | 7 |
| Marjorie | 5 | 9 |

42 sec with an average of 4 min 14 sec. Each elephant made between 133 and 277 sensors' grasps, including 26 to 112 pinches.

## Data analyses

For each session, the collected and time-stamped data are compiled in a table showing the force exerted on the two sensors. Analysis of this table alongside the videos of the experiment enabled us to note, for each grasp, the identity of the elephant, the maximum force on each sensor, the type of grasp, i.e. pinch, wrap or other, the sensors' orientation, i.e. vertical or horizontal, the bending of the trunk, i.e. unbent or bent, and the association between fingers and sensors, i.e. which finger is placed on which sensor. The maximum pinch force across all sessions was then recorded in a table for each elephant depending on the sensors' orientation and bending of the trunk. To compare the force according to the sensors' orientation, the data has been made comparable and the individual effect has been removed. For each comparison, only the elephants that performed pinches in both orientations were kept in the analysis. In addition, so that the data from each elephant had the same weight in the analysis, the number of force measurements considered was that of the elephant that performed the fewest pinches (Table 3). For example, if two elephants A and B pinched the sensors 6 and 10 times respectively, the comparison will be made with the 6 force measurements of elephant A and the 6 highest force measurements of elephant B. Then, boxplots were carried out with weighted data using the R version 4.1.1 software [30] and the "ggboxplot" function of the "ggpubr" package [31]. Wilcoxon signed-rank tests were used to compare the average force of the pinches on

**Table 3. Number of elephants and pinches observed according to the analysis.**

| Analysis | Trunk position | sensors' orientation | Fig number | Number of elephants | Number of pinches observed per elephant |
|---|---|---|---|---|---|
| Comparisons of pinching force according to the sensors' orientation | Unbent | Vertical | 4 | 4 | 7 |
|  |  | Horizontal |  |  | 7 |
|  | Bent | Vertical | 5 | 2 | 29 |
|  |  | Horizontal |  |  | 29 |
| Comparisons of pinching force according to the trunk position | Unbent | Vertical | No Fig | 3 | 4 |
|  | Bent |  |  |  | 4 |
|  | Unbent | Horizontal | No Fig | 1 | 25 |
|  | Bent |  |  |  | 25 |
| Comparisons of the difference in force between the dorsal and ventral fingers | Unbent | Vertical | 6 | 5 | 162 |
|  |  | Horizontal |  | 4 | 75 |
|  | Bent | Vertical |  | 3 | 73 |
|  |  | Horizontal |  | 3 | 68 |

The raw data are available in the repository Open Science Framework [33]

vertical and horizontal sensors using the "wilcox.test" of the "stats" package [30]. This analysis has already been carried out in other studies, for example for maximum bite force comparisons [32]. The pinch forces are considered to be statistically different when the p-value is strictly less than 0.05. To compare the force performed by the two trunk fingers, for each sensors' orientation and bending of the trunk, the difference in force between the two fingers of the proboscis was obtained by subtracting the force of the dorsal finger from that of the ventral finger for each pinch. Boxplots and Wilcoxon signed-rank tests were carried out, using the same R packages as previously mentioned. The dots above the 0 line represent a pinch where the dorsal finger was stronger, and the dots below the 0 line represent a pinch where the ventral finger was stronger.

## Results

### The trunk positions during pinches

The elephants pinched on the sensors using four trunk positions. Indeed, with the vertical sensors, the elephants positioned their trunks either straight out in front of them (Fig 3A) or with

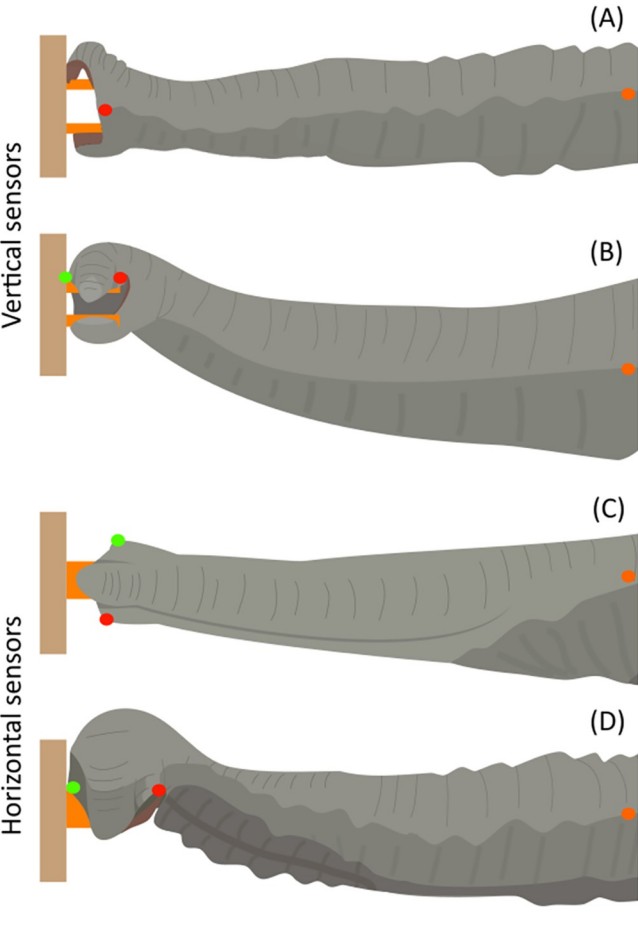

**Fig 3. Trunk positions during the pinching of the force sensors, from the side view.** The two first pictures represent pinches on the vertical sensors when the trunk was straight (A) and bent (B). The two-second pictures illustrate pinches on the horizontal sensors when the trunk was twisted (C) and twisted and bent (D). The coloured dots represent anatomical landmarks to help understand the movements of the trunk on the drawing.

the distal end of the trunk bent (Fig 3B). With the horizontal sensors, the elephants positioned their trunks either twisted (Fig 3C) or twisted and bent (Fig 3D). The number of pinches carried out in each trunk position have been summarised in Table 3.

### The maximum trunk tip force during pinches

**Unbent trunk.** For all elephants combined, the maximum pinching force with the vertical sensors, i.e. the trunk straight, was 86.4 N and 56.3 N with the horizontal sensors, i.e. the trunk twisted (S1 Table).

This experiment enabled us to compare the average force of pinches on vertical and horizontal sensors. The results showed that the pinches on vertical sensors were stronger on average (53.53 N) than the ones on horizontal sensors (32.62 N) (W = 176, p-value = 0.0002776; Fig 4).

**Bent trunk.** For all elephants combined, the maximum pinching force with the vertical sensors, i.e. the trunk bent, was 67.7 N and 66.7 N with the horizontal sensors, i.e. the trunk bent and twisted (S1 Table).

When the trunk was slightly bent during pinching, in the same way as when the trunk was straight, the pinches on vertical sensors were stronger on average (38.66 N) than the ones on horizontal sensors (32.37N) (W = 1129, p-value = 0.002283). However, the difference in force

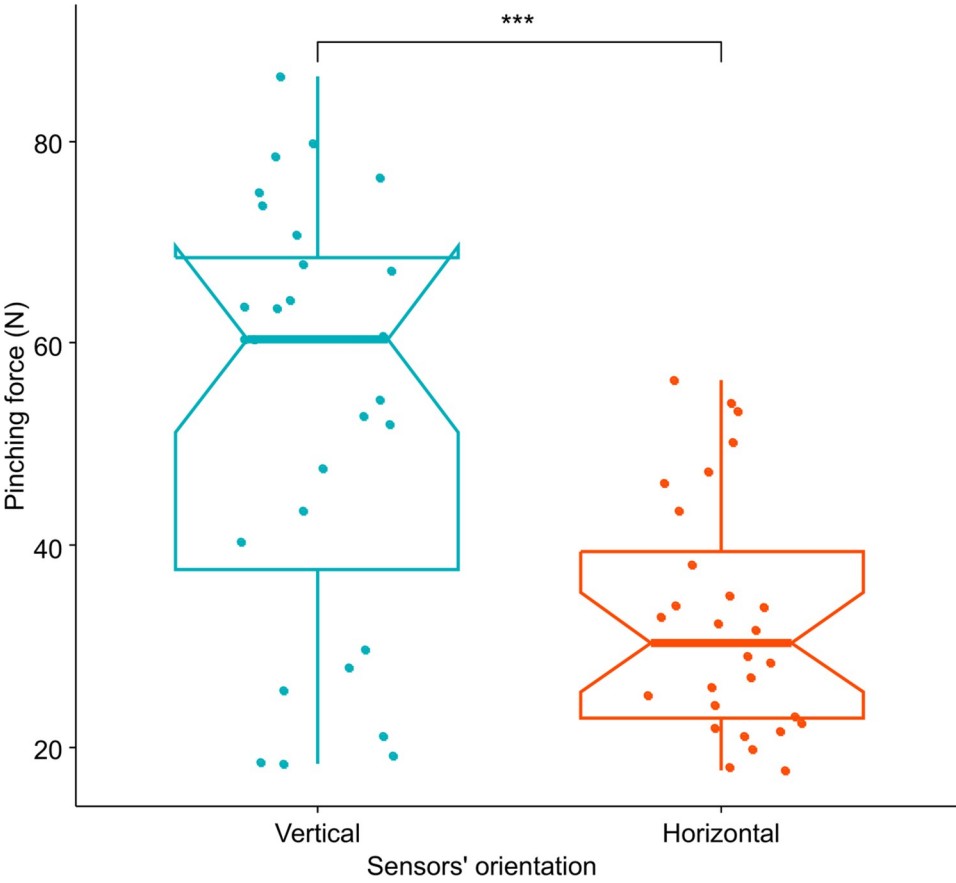

**Fig 4. Comparison of pinching force according to the sensors' orientation when the trunk was straight.** The force is expressed in Newton (N). Each colour represents a sensors' orientation: blue for pinches on vertical sensors and red for pinches on horizontal sensors. The p-value of the Wilcoxon signed-rank tests used to compare sensors' orientation, less than 0.001, was summarized with three asterisks.

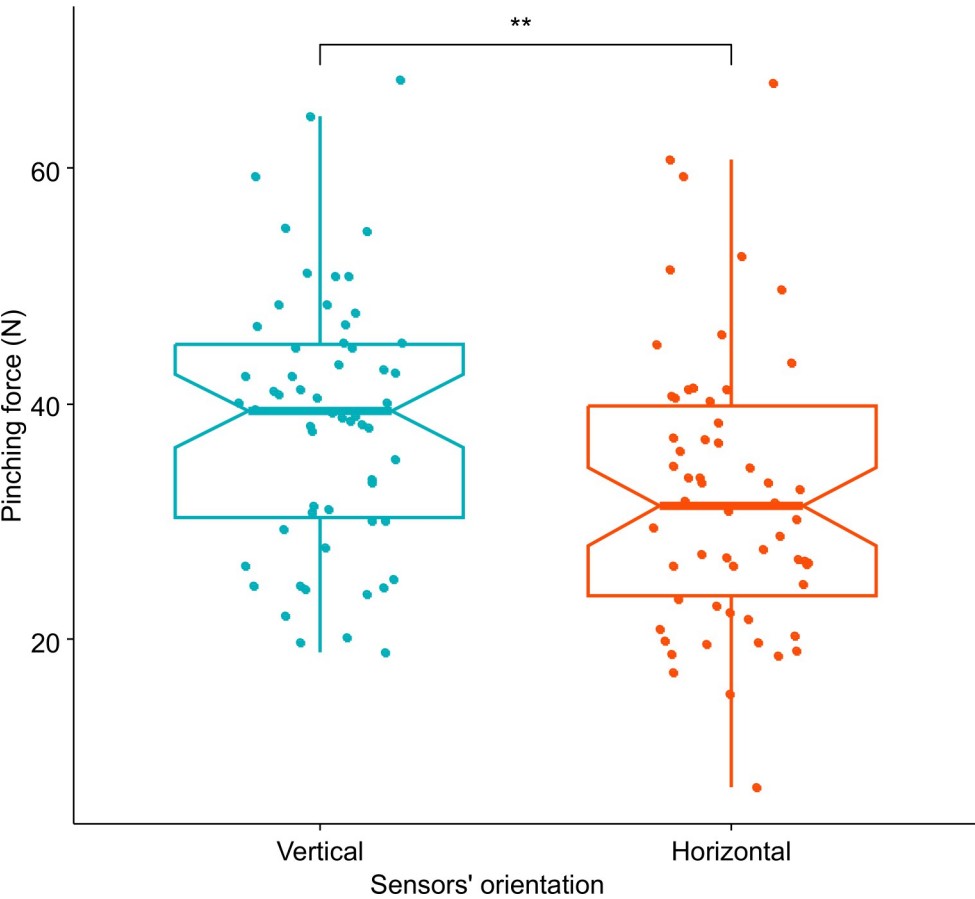

**Fig 5. Comparison of pinching force according to the sensors' orientation when the trunk was bent.** The force is expressed in Newton (N). Each colour represents a sensors' orientation: blue for pinches on vertical sensors and red for pinches on horizontal sensors. The p-value of the Wilcoxon signed-rank tests used to compare sensors' orientation, less than 0.01, was summarized with two asterisks.

between the two sensors' orientation was greater when the trunk was bent than when the trunk was bent and twisted (Fig 5).

**Comparison of pinch force between unbent and bent trunk positions.** A comparison of the average pinch force as a function of the bending of the trunk showed no difference between the unbent and bent trunk forces, regardless of the sensors' orientation (vertical sensors: unbent = 35.06 N vs bent = 38.65 N, Wilcoxon signed-rank tests: W = 88, p-value = 0.3777 and horizontal sensors: unbent = 26.19 N vs bent = 25.08 N, Wilcoxon signed-rank tests: W = 317, p-value = 0.9387). However, the results also showed that the maximum pinch force on vertical sensors was higher when the trunk was unbent (67.7 N bent vs 86.3 N unbent), while the maximum pinch force on horizontal sensors was higher when the trunk was bent (66.7 N bent vs 55.9 N unbent; S1 Table).

## Force distribution between the fingers

We analysed the distribution of force between the two fingers of the trunk as a function of the position of the trunk.

**Unbent trunk.** The results showed that the two fingers of the trunk did not apply the same force, regardless of the sensors' orientation. On average, the dorsal finger pressed harder

on the sensor than the ventral finger during a pinch on vertical (dorsal: 30.21 N vs ventral: 8.83 N) or horizontal sensors (dorsal: 12.07 N vs ventral: 10.30 N). However, no pinch where the ventral finger was stronger than the dorsal finger was recorded with the vertical sensors, whereas many pinches were recorded where the ventral finger was stronger than the dorsal finger with vertical sensors (Fig 6). In addition, considering only maximum force, the dorsal finger could apply up to 60.2 N during a pinch on vertical sensors and 34.4 N during a pinch on horizontal sensors, while the ventral finger could apply up to 30.1 N during a pinch on vertical sensors and 38.1 N during a pinch on horizontal sensors (S2 Table). Finally, the difference in force between the dorsal and ventral fingers was higher during pinches on vertical sensors than during pinches on horizontal sensors (W = 11516, p-value < 2.2e-16; Fig 6).

**Trunk bent.** The dorsal finger always pressed harder on the vertical sensor than the ventral finger during pinches (on average, dorsal: 28.74 N vs ventral: 6.67 N; Fig 6). Actually, the dorsal finger could apply up to 58 N, while the ventral finger could apply up to 18.6 N (S2 Table). On the contrary, the ventral finger pressed on average harder on the sensor than the dorsal finger and most pinches were recorded with the ventral finger being stronger than the dorsal finger on horizontal sensors (dorsal: 12.75 N vs ventral: 17.46 N; Fig 6). The dorsal finger could apply up to 37.6 N, while the ventral finger could apply up to 41.7 N (S2 Table). In addition, as with the trunk unbent, the difference in force between the dorsal and ventral fingers was higher during pinches on vertical sensors than during pinches on horizontal sensors (W = 4767, p-value < 2.2e-16; Fig 6).

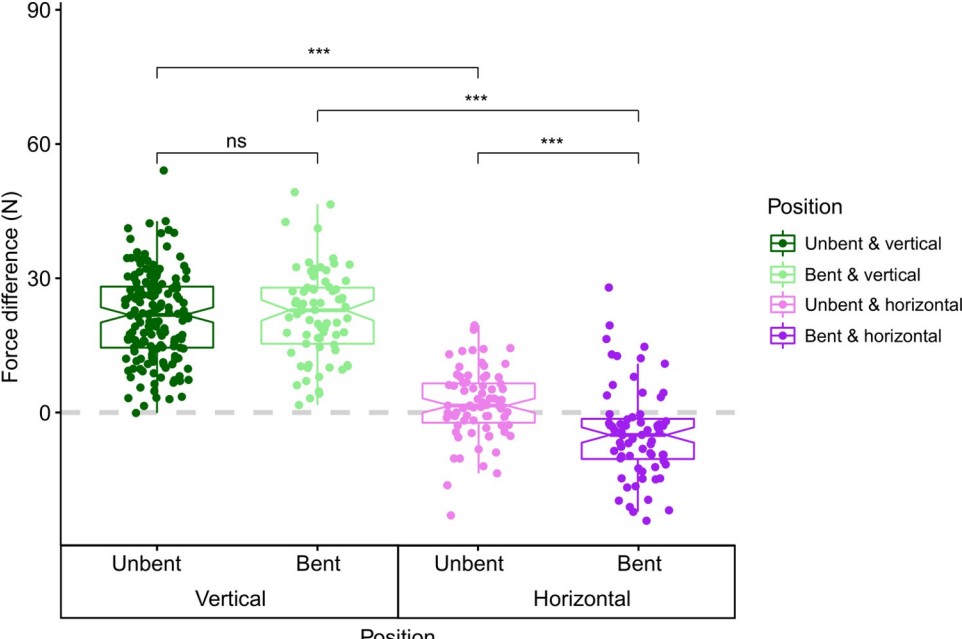

**Fig 6. Comparison of the difference in force between the dorsal and ventral fingers according to the trunk position.** The dots above the 0 line represent a pinch where the dorsal finger was the stronger of the two and the dots below the 0 line represent a pinch where the ventral finger was the stronger of the two. The force is expressed in Newton (N). Each colour represents a combination of trunk position and sensor orientation: dark green for the unbent trunk with vertical sensors, light green for the bent trunk with vertical sensors, dark violet for the unbent trunk with horizontal sensors, and light violet for the bent trunk with horizontal sensors. P-values of the Wilcoxon signed-rank tests used to difference in force between the fingers were summarized with asterisks: "ns" means p-value > 0.05 and "***" means p-value ≤ 0.001.

**Comparison of fingers' force difference between unbent and bent trunk positions.** The difference in force between the dorsal and ventral fingers was similar whatever the trunk position during pinching on vertical sensors (W = 6095, p-value = 0.7067).

On the contrary, this difference increased when the trunk was unbent compared to when it was bent during pinching on horizontal sensors (W = 1390.5, p-value = 2.8e-06; Fig 6).

## Discussion

### The maximum trunk tip force during pinches

This work, the first with a headcount of 5 elephants, showed that the female African savannah elephants can exert a trunk tip force at least of 86.3 N, which is consistent with our hypothesis of a force between 32.79 N and 257 N. We know that a female elephant can lift at least 63 kg (~618 N) by wrapping her trunk around a dumbbell [7]. However, the nostrils dominate the internal volume at the trunk tip, thus the muscles have less space which may explain the lower force recorded by our study. Indeed, only four of the six muscle groups are present in the distal part of the trunk. The dorsal tip finger is a continuation of the dorsal part of the trunk, with radial and longitudinal muscles extending almost to the very distal end. On the ventral side, superficial oblique muscles become more longitudinally aligned close to the tip, whereas deep oblique muscle fibres seem to gradually align and merge with the radial ventral muscles [2]. In addition, the prehensile trunk tip is used to grasp small objects with high precision that does not require great force [2, 8, 10, 11], so it seems to appear a functional division between the "body" and fingers of the trunk, which are associated with the powerful and the precision respectively.

### Variability of the trunk tip force according to the trunk position

The results show that the pinching force was similar on average between the straight and bent trunk, regardless of the sensors' orientation, disproving our hypothesis. We assumed that the formation of a joint could increase the pinching force, such as for the compressing force [8]. It would appear that the bend did not allow the pinch force to be increased. However, the results showed that some elephants were more comfortable bending their trunk to deploy their maximum force, while others were more comfortable with their trunk straight to force. These results show an inter-individual variation in the effect of trunk position on the value of maximum force. It would be interesting to study the variation in force and preferred trunk positions between individuals as a function of their morphological characteristics.

Unlike bending, torsion of the trunk was related to the force measurement. Regardless of the trunk position, the pinching force on vertical sensors was higher than the one on horizontal sensors. Each movement is the result of simultaneous contractions and releases of different muscles. The torsion is possible thanks to the contraction of the superficial and deep oblique muscles located in the ventral part of the trunk. The stress generated by the ventral oblique muscles can be transmitted to the dorsal side by the connective tissue wrapped around the muscular core [2]. The trunk muscles also produce the force the elephant needs to grasp objects. The inward bending of the dorsal and ventral fingers, necessary for pinching, is caused by the contraction of the radial groups, causing a passive elongation of the outer longitudinal fibres, and the merged deep oblique and radial ventral muscles respectively [2]. We can therefore assume the existence of a compromise between torsion movement and pinch force in using the trunk muscles, which could explain a reduction in pinch force during a torsion movement.

## Force distribution between the fingers

The dorsal and ventral fingers reached maximum forces of 60.2 N and 41.7 N respectively. The dorsal finger was on average stronger than the ventral finger. These results are consistent with our hypothesis linked to the difference in relative volume between the muscles in the two fingers. The muscles present in the dorsal part represent 56% of the relative volume of the muscles of the trunk, whereas those present in the ventral part represent 37% [2]. In addition, the different geometry of the trunk wrinkles and folds, which influence its elasticity, and the different oblique muscle proportion between the two trunk parts, allow the dorsal part of the trunk tip to stretch 5.1 cm further than the ventral part [34]. The skin on the dorsal part of the trunk comprises a series of folds whose length is greater than that of the ventral region. Moreover, in the ventral-distal part, the oblique muscles represent the largest proportion of the cross-section and may limit the trunk extension [2, 34]. The dorsal finger could therefore have a better grip on the sensors, which could explain the difference in force.

In comparison, in humans, when the thumb and index finger pinch, the forces are not always equal. Depending on the surface, the thumb may be stronger than the index finger (thumb: 7.48±0.13 N vs index: 6.11±0.20 N), or both fingers may be equally strong (thumb: 6.32±0.14 N vs index: 6.08±0.14 N) [35]. In addition, the directions of force vectors were dependent on the magnitude of the pinch force applied, especially at force levels below 2 N. The two force vectors were never observed to be perfectly opposed, whereby not intrinsically satisfying static equilibrium [36]. The same phenomenon could occur between the two fingers of the elephant's trunk, explaining the greater force of the dorsal finger compared to the ventral one.

However, the force distribution between the fingers was related to the trunk position. On one hand, torsion reduced the difference in force between the fingers and increased the number of pinches in which the ventral finger was the strongest. Indeed, the torsion reduced the dorsal finger's maximum force and increased the ventral finger's maximum force. On the other hand, bending reduced the maximum force of the dorsal and ventral fingers and the difference in force between the fingers was similar on average between the unbent and bent trunk with the vertical sensors. With the horizontal sensors, the bending, combined with torsion, increased the difference in force between the fingers and the number of pinches in which the ventral finger was the strongest. The bending increased the maximum force of the dorsal and ventral fingers but not in the same proportions. The muscles used to bend the trunk and to pinch are the same, i.e. the longitudinal and radial groups. The oblique muscles are used to twist the trunk and are also present in the ventral finger, more aligned longitudinally than in the rest of the trunk. It is possible that the contraction of these muscles, which twist the trunk, indirectly increases the force of the ventral finger.

Torsion of the trunk by contraction of the oblique muscles will twist the longitudinal muscle fibres, also affecting the length of the trunk [5]. This deformation of the longitudinal muscles may prevent them from contracting correctly and thus affects the ability of the dorsal finger to bend to perform the pinch. The radial muscle fibres, lying perpendicular to the longitudinal, are less affected by torsion. Thus, the force of the dorsal finger decreases and that of the ventral finger increases during twisting. It could be interesting for future robots to have oblique, longitudinal and radial 'muscles' in symmetry on the ventral and dorsal sides to avoid certain movements reducing the pinch force.

Desirable properties of soft robot actuators include infinite degree of freedom, large deformation, high compliance and relatively large output force [for a review on soft robotic grippers see 37]. Currently, the reliability of gripping small objects is often not high due to the small contact area [38]. The reliability of pinch gripping has been improved by creating a two-finger

soft-robotic gripper with modules of different heights [38]. However, their gripper and all other existing two-finger soft-robotic grippers have a symmetrical constitution with fingers of the same shape and deploying the same force [examples: 38–40]. There are already asymmetrical robots mimicking the trunk body, similar to the non-homogeneity of the elephant trunk during stretching, which is not due to muscle but to asymmetries in the properties of the skin [14, 34, 41]. This study also highlights the benefits of asymmetries in terms of shape and force between the two fingers, which could lead to more versatile robotic grippers that are efficient gripping without having to deploy greater force.

## Conclusion

In this study, we measured that the maximum pinch force of the trunk of female African savanna elephants was at least 86.4 N and deduced that this part of the trunk was mainly dedicated to precision grasping. We have highlighted the difference in force between the two fingers of the trunk, with the dorsal finger predominantly stronger than the ventral finger, and finally, we have shown that the position of the trunk, and particularly the torsion, influences its force but also the distribution of force between the two fingers of the trunk. The difference in force between the elephant's two fingers has never been studied before, and we hope that these results will open up new food for thought about soft grippers.

## Supporting information

**S1 Fig. Maximum pinch force of the trunk tip with vertical sensors by session when the trunk is unbent.** The force is expressed in Newton (N). Each colour represents an elephant: orange for Ashanti, grey for Juba, dark blue for M'Kali, yellow for Marjorie and light blue for Tana.
(TIF)

**S2 Fig. Maximum pinch force of the trunk tip with horizontal sensors by session when the trunk is unbent.** The force is expressed in Newton (N). Each colour represents an elephant: orange for Ashanti, grey for Juba, yellow for Marjorie and light blue for Tana.
(TIF)

**S3 Fig. Maximum pinch force of the trunk tip on vertical sensors by session when the trunk is bent.** The force is expressed in Newton (N). Each colour represents an elephant: dark blue for M'Kali, yellow for Marjorie and light blue for Tana.
(TIF)

**S4 Fig. Maximum pinch force of the trunk tip on horizontal sensors by session when the trunk is bent.** The force is expressed in Newton (N). Each colour represents an elephant: orange for Ashanti, dark blue for M'Kali and yellow for Marjorie.
(TIF)

**S5 Fig. Photo of the fake pinch sensors.**
(TIF)

**S1 Table. Maximum pinch force of the trunk tip per elephant on vertical and horizontal sensors with the trunk unbent or bent.**
(DOCX)

**S2 Table. Maximum force of the trunk finger pinch on vertical and horizontal sensors with the trunk unbent or bent.**
(DOCX)

**S3 Table. The number of pinches and total grasps by elephants according to the orientation of the sensors.**
(DOCX)

## Acknowledgments

We warmly thank the ZooParc de Beauval for giving us access to its elephants and welcoming our project. We would like to thank in particular the "elephant team" under the supervision of Matthieu Villemain: Johanna Neto, Amaury Boutier, Priscille M. d'Annoville, Matthieu Fromet, Laura Rastrelli, Mathieu Hysebergue, Clément Langlès and Valentin Thénault for their competences and professionalism. We thank Guillaume Joubert for his recommendations and help in improving the Arduino code. Finally, we are grateful to the two reviewers who provided helpful and constructive reviews that improved the manuscript.

## Author Contributions

**Conceptualization:** Pauline Costes, Emmanuelle Pouydebat, Raphaël Cornette.

**Formal analysis:** Pauline Costes.

**Funding acquisition:** Baptiste Mulot, Emmanuelle Pouydebat.

**Methodology:** Pauline Costes, Arnaud Delapré, Céline Houssin.

**Resources:** Baptiste Mulot.

**Supervision:** Emmanuelle Pouydebat, Raphaël Cornette.

**Writing – original draft:** Pauline Costes.

**Writing – review & editing:** Pauline Costes, Arnaud Delapré, Céline Houssin, Baptiste Mulot, Emmanuelle Pouydebat, Raphaël Cornette.

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
