## [Decision Letter · Decision Letter 0]

5 Dec 2023

PONE-D-23-31488Maximum trunk tip force assessment related to trunk position and prehensile 'fingers' implication in African savannah elephantsPLOS ONE

Dear Dr. Costes,

Thank you for submitting your manuscript to PLOS ONE. After careful consideration, we feel that it has merit but does not fully meet PLOS ONE’s publication criteria as it currently stands. Therefore, we invite you to submit a revised version of the manuscript that addresses the points raised during the review process.

We look forward to receiving your revised manuscript.

Kind regards,

Monika Błaszczyszyn

Academic Editor

PLOS ONE

3.We note that the grant information you provided in the ‘Funding Information’ and ‘Financial Disclosure’ sections do not match.

4. We note that Figure 1 in your submission contain copyrighted images. All PLOS content is published under the Creative Commons Attribution License (CC BY 4.0), which means that the manuscript, images, and Supporting Information files will be freely available online, and any third party is permitted to access, download, copy, distribute, and use these materials in any way, even commercially, with proper attribution. For more information, see our copyright guidelines: http://journals.plos.org/plosone/s/licenses-and-copyright.

5. We notice that your supplementary figures are included in the manuscript file. Please remove them and upload them with the file type 'Supporting Information'. Please ensure that each Supporting Information file has a legend listed in the manuscript after the references list.

Reviewers' comments:

Reviewer's Responses to Questions

**Comments to the Author**

1. Is the manuscript technically sound, and do the data support the conclusions?

Reviewer #1: Partly

Reviewer #2: Partly

2. Has the statistical analysis been performed appropriately and rigorously? 

Reviewer #1: Yes

Reviewer #2: Yes

3. Have the authors made all data underlying the findings in their manuscript fully available?

Reviewer #1: No

Reviewer #2: Yes

4. Is the manuscript presented in an intelligible fashion and written in standard English?

Reviewer #1: Yes

Reviewer #2: Yes

5. Review Comments to the Author

Reviewer #1: Strong aspects:

The authors measure the force of the fingers of the trunk of several elephants and find out that one finger has stronger force than the other.

Weak aspects:

Although the findings are new the possible implications or applications of this particularity of the elephant trunk are not mentioned. I suggest to improve of the paper by giving possible reasons why the dorsal finger is more powerful. Also, a comparison with the human hand would be interesting.

Comments

The latin name for this specie of elephant is given twice: at the beginning of the paper and around line 200.

Reviewer #2: This manuscript measures the distal gripping force of female African savanna elephants through a cleverly designed experiment on captive individuals. The study is able to display interesting results related to maximum pinch force, as well as differences between the two fingers of the trunk. I enjoyed reading the manuscript but I do think that it requires more polishing before acceptance. Please see my comments below. I also suggest that the manuscript is proof read for the English, as some sections were either a little difficult to make sense of, or words such as "Indeed" are over-used. I hope that my comments will be helpful in improving your manuscript. Well done.

Please be careful with your changing of tenses within sections. I have pointed it out a couple times in the Methods but I see it happening in the Results as well.

Introduction

Page 3, line 48: Scientific name for Asian elephants can be mentioned earlier in the Intro when you first mention them.

Methods

Page 5, line 101: Change to "and then in April 2023."

Page 5, line 102: no need to re-mention the scientific name here.

Table 1: Rather add a column heading for birth year and a column heading for birth location. Then the Origin column can be divided into these 2 columns to make for easier reading.

Page 5, line 106: Rather say "In this study, we observed elephants in an indoor setting."

Page 5, lines 106-108: Please rephrase the English of the second sentence of this paragraph.

Page 6, lines 118-127: Stick to the same tense for the paragraph. Here it changes from present to past tense. The same goes for the "The Course of the Experiment" paragraph.

Page 6, line 122: Change to "pinch hard enough to pass..."

Page 8, lines 180-183: Please combine these sentences with a comma.

Page 9, line 192: Rather say "as previously mentioned."

Table 2: I think that Table 2 can be made more clear to the reader. Look at splitting the 3rd and 4th columns into subheadings where the value is placed below. The current state makes it difficult to understand and appreciate the variation of the recorded values.

Results

Page 10, line 204: Rather say "summarised" instead of "summed up"

Page 11, lines 216-218: Even though the p values are shown in the graph, please provide the full statistics here and the average force values.

Page 11, lines 228-229: Give the average values if you can here.

Page 11, line 230: What do you mean more important? Do you mean greater?

Page 12, lines 241-244: Here in brackets you can actually put what the maximum bent and unbent values were. E.g. - (??N bent vs ??N unbent).

Figure 6 - Instead of placing the full p-values for all of your figures, rather use the *, **, *** code system representing p<0.05; 0.01; 0.001. It will help neaten the figures.

Page 13, lines 272-273: Please correct the English of this sentence.

Page 14, line 285: Again, rather rephrase the word "important" to something of increased or decreased

Discussion

Page 14, line 294: The lower force recorded by your study? Please be specific.

Page 14, line 299: Remove "we know" - the audience may not know.

Page 14, lines 304-305: Remove "On one hand" and then instead of saying "Our hypothesis was wrong", rather add "disproving our hypothesis" to the end of the first sentence.

Page 15, line 313: Remove "On the other hand"

Page 15, line 319: The word "Indeed" is used a lot in the Discussion. Try substitute it for other words of the same meaning.

Page 15, line 327: Remove "All pinches combined"

Page 15, line 328: Change is to was

Page 16, lines 399-355: Please try and rewrite this paragraph in a more concise manner. It feels very long-winded and I am sure could be reduced into a more concise description.

I noticed that you spoke about robotics extensively in the Introduction, which lead me to believe that the Discussion would loop back to this, however, robotics is never mentioned in the Discussion. I suggest either cutting down its portion in the Intro drastically, or show here in the Discussion how what you have found relates back to robotics.

6. PLOS authors have the option to publish the peer review history of their article (what does this mean?). If published, this will include your full peer review and any attached files.

Reviewer #1: No

Reviewer #2: No

---

## [Author Response · Author response to Decision Letter 0]

24 Jan 2024

January 22nd, 2023

Dear Academic Editor of PLOS ONE,

We would like to thank the reviewers for their time and precious comments on our manuscript. We have edited the paper in the light of their remarks.

We hope that this corrected manuscript is now suitable for publication.

On behalf of all authors.

Pauline Costes

PhD Student at the Muséum National d'Histoire Naturelle

Academic Editor

Dear Dr. Costes,

Thank you for submitting your manuscript to PLOS ONE. After careful consideration, we feel that it has merit but does not fully meet PLOS ONE’s publication criteria as it currently stands. Therefore, we invite you to submit a revised version of the manuscript that addresses the points raised during the review process.

We look forward to receiving your revised manuscript.

Kind regards,

Monika Błaszczyszyn

Academic Editor

PLOS ONE

→ Thank you very much for this feedback on our paper. We hope this corrected version is now suitable for publication in PLOS ONE.

→ Thank you for this reminder, our manuscript meets PLOS ONE's style requirements.

→ We have deposited the raw data in the repository Open Science Framework. They are now freely available by following this link: https://osf.io/85kj3/?view_only=d245827940504ad7b6f0461de2b1301b

We have also added the reference in the appropriate section. Costes, P., Delapré, A., Houssin, C., Mulot, B., Pouydebat, E., & Raphaël, C. (2023, December 7). Elephant pinching forces. Retrieved from osf.io/85kj3

→ Thank you for pointing out this error. We have updated our declaration for the 'Financial Disclosure' section in the cover letter and the 'Funding Information' section in the submission form.

We would like to make the following changes to our financial statement in the 'Financial Disclosure' section: « This work was supported by grants IBEES 20SB403U7209, Association Sorbonne Université (EP) and CNRS 80 PRIME, Centre National de la Recherche Scientifique (EP), which we thank. The funders had no role in study design, data collection and analysis, decision to publish, or preparation of the manuscript. »

4. We note that Figure 1 in your submission contain copyrighted images. All PLOS content is published under the Creative Commons Attribution License (CC BY 4.0), which means that the manuscript, images, and Supporting Information files will be freely available online, and any third party is permitted to access, download, copy, distribute, and use these materials in any way, even commercially, with proper attribution. For more information, see our copyright guidelines: http://journals.plos.org/plosone/s/licenses-and-copyright.

please specify in the figure caption text when a figure is similar but not identical to the original image and is therefore for illustrative purposes only.

→ We realise that we have completed the form incorrectly during submission. Our Figure 1 has been created from scratch by the authors and the represented device is not the same. Only the design is similar (but not identical) to the original image. However, if authorisation is still required, we can make the request. The apples in this figure were drawn by the authors.

4bis. Thank you for clarifying the creation of this image. Could you please clarify whether the source of the apple art in this figure specifically are author-generated or whether they are externally sourced? If so, please provide the source of this image specifically.

→ The apples in this figure were drawn by the authors.

5. We notice that your supplementary figures are included in the manuscript file. Please remove them and upload them with the file type 'Supporting Information'. Please ensure that each Supporting Information file has a legend listed in the manuscript after the references list.

→ Thank you for this clarification. We have removed the supplementary figures in the manuscript file and uploaded them with the file type 'Supporting Information'.

Reviewer #1: Strong aspects:

The authors measure the force of the fingers of the trunk of several elephants and find out that one finger has stronger force than the other.

Weak aspects:

Although the findings are new the possible implications or applications of this particularity of the elephant trunk are not mentioned. I suggest to improve of the paper by giving possible reasons why the dorsal finger is more powerful. Also, a comparison with the human hand would be interesting.

→ Thank you for your comments. As you advised, we discussed the implications of the difference in force between the two trunk fingers and added a comparison with the human hand which allows us to discuss another explanation for the difference in force between the two fingers, in addition to the morphological and anatomical differences already mentioned.

The following paragraph has been added to the discussion in the "Force distribution between the fingers" section (lines 342): “In comparison, in humans, when the thumb and index finger pinch, the forces are not always equal. Depending on the surface, the thumb may be stronger than the index finger (thumb: 7.48±0.13 N vs index: 6.11±0.20 N), or both fingers may be equally strong (thumb: 6.32±0.14 N vs index: 6.08±0.14 N) (Dollahon et al, 2022). In addition, the directions of force vectors were dependent on the magnitude of the pinch force applied, especially at force levels below 2 N. The two force vectors were never observed to be perfectly opposed, whereby not intrinsically satisfying static equilibrium (Li et al., 2013). The same phenomenon could occur between the two fingers of the elephant's trunk, explaining the greater force of the dorsal finger compared to the ventral one.”

Dollahon, Devon, Seokchang Ryu, et Hangue Park. « Pinching Force Changes by Modulating the Interaction Gain Over the Fingertip ». IEEE Access 10 (2022): 9744 49. https://doi.org/10.1109/ACCESS.2022.3143837.

Li, Ke, Raviraj Nataraj, Tamara L. Marquardt, et Zong-Ming Li. « Directional Coordination of Thumb and Finger Forces during Precision Pinch ». PLOS ONE 8, no 11 (2013): e79400. https://doi.org/10.1371/journal.pone.0079400.

Comments

The latin name for this specie of elephant is given twice: at the beginning of the paper and around line 200.

→ Thank you for your comment, the second mention of "Loxodonta africana" on line 102 has been deleted.

Reviewer #2: This manuscript measures the distal gripping force of female African savanna elephants through a cleverly designed experiment on captive individuals. The study is able to display interesting results related to maximum pinch force, as well as differences between the two fingers of the trunk. I enjoyed reading the manuscript but I do think that it requires more polishing before acceptance. Please see my comments below. I also suggest that the manuscript is proof read for the English, as some sections were either a little difficult to make sense of, or words such as "Indeed" are over-used. I hope that my comments will be helpful in improving your manuscript. Well done.

Please be careful with your changing of tenses within sections. I have pointed it out a couple times in the Methods but I see it happening in the Results as well.

→ Thanks for the positive feedback, we've improved the English of this manuscript, notably by using the word "Indeed" less and by harmonizing the tenses used.

Introduction

Page 3, line 48: Scientific name for Asian elephants can be mentioned earlier in the Intro when you first mention them.

→ The mention of "Elephas maximus" on line 48 has been moved to line 40.

Methods

Page 5, line 101: Change to "and then in April 2023."

→ The sentence has been modified, thank you.

Page 5, line 102: no need to re-mention the scientific name here.

→ Thank you for your comment, the second mention of "Loxodonta africana" on line 102 has been deleted.

Table 1: Rather add a column heading for birth year and a column heading for birth location. Then the Origin column can be divided into these 2 columns to make for easier reading.

→ The "Origin" column has been split in two to create two separate columns for "Birth location" and "Birth year". The table is now easier to read.

Page 5, line 106: Rather say "In this study, we observed elephants in an indoor setting."

→ The sentence has been modified, thank you.

Page 5, lines 106-108: Please rephrase the English of the second sentence of this paragraph.

→ The sentence “Every morning, one to three elephants isolated were groomed and/or trained to facilitate any future medical interventions in a pen dedicated to daily care and medical training.” was replaced by the following: " Every morning, time is devoted to medical training: one to three elephants are isolated in an enclosure dedicated to daily care and grooming, to facilitate any future medical intervention."

Page 6, lines 118-127: Stick to the same tense for the paragraph. Here it changes from present to past tense. The same goes for the "The Course of the Experiment" paragraph.

→ All the verbs in the paragraphs "Functioning of the device" and "The Course of the experiment" are now in the past tense.

Page 6, line 122: Change to "pinch hard enough to pass..."

→ The sentence has been modified, thank you.

Page 8, lines 180-183: Please combine these sentences with a comma.

→ The sentences have been combined to become: “For example, if two elephants A and B pinched the sensors 6 and 10 times respectively, the comparison will be made with the 6 force measurements of elephant A and the 6 highest force measurements of elephant B.”

Page 9, line 192: Rather say "as previously mentioned."

→ The sentence has been modified, thank you.

Table 2: I think that Table 2 can be made more clear to the reader. Look at splitting the 3rd and 4th columns into subheadings where the value is placed below. The current state makes it difficult to understand and appreciate the variation of the recorded values.

→ Table 2 has been clarified and we hope it will be easier to read.

Results

Page 10, line 204: Rather say "summarised" instead of "summed up"

→ The sentence has been modified, thank you.

Page 11, lines 216-218: Even though the p values are shown in the graph, please provide the full statistics here and the average force values.

Page 11, lines 228-229: Give the average values if you can here.

→ Thank you for pointing out these omissions, we have provided the full statistics for each test and all the mean values mentioned in the results section.

Page 11, line 230: What do you mean more important? Do you mean greater?

→ Thank you for pointing out that imprecision, we did mean greater. The sentence has been modified.

Page 12, lines 241-244: Here in brackets you can actually put what the maximum bent and unbent values were. E.g. - (??N bent vs ??N unbent).

→ The maximum force values have been added to the sentence. The new sentence is as follows: “However, the results also showed that the maximum pinch force on vertical sensors was higher when the trunk was unbent (67.7 N bent vs 86.3 N unbent), while the maximum pinch force on horizontal sensors was h

---

## [Decision Letter · Decision Letter 1]

18 Mar 2024

Maximum trunk tip force assessment related to trunk position and prehensile fingers implication in African savannah elephants

PONE-D-23-31488R1

Dear Dr. Costes,

We’re pleased to inform you that your manuscript has been judged scientifically suitable for publication and will be formally accepted for publication once it meets all outstanding technical requirements.

An invoice for payment will follow shortly after the formal acceptance. To ensure an efficient process, please log into Editorial Manager at Editorial Manager® , click the 'Update My Information' link at the top of the page, and double check that your user information is up-to-date. If you have any billing related questions, please contact our Author Billing department directly at authorbilling@plos.org.

Kind regards,

Monika Błaszczyszyn

Academic Editor

PLOS ONE

Additional Editor Comments (optional):

Reviewers' comments:

Reviewer's Responses to Questions

**Comments to the Author**

1. If the authors have adequately addressed your comments raised in a previous round of review and you feel that this manuscript is now acceptable for publication, you may indicate that here to bypass the “Comments to the Author” section, enter your conflict of interest statement in the “Confidential to Editor” section, and submit your "Accept" recommendation.

Reviewer #1: All comments have been addressed

Reviewer #2: All comments have been addressed

2. Is the manuscript technically sound, and do the data support the conclusions?

Reviewer #1: Yes

Reviewer #2: Yes

3. Has the statistical analysis been performed appropriately and rigorously? 

Reviewer #1: Yes

Reviewer #2: Yes

4. Have the authors made all data underlying the findings in their manuscript fully available?

Reviewer #1: Yes

Reviewer #2: Yes

5. Is the manuscript presented in an intelligible fashion and written in standard English?

Reviewer #1: Yes

Reviewer #2: Yes

6. Review Comments to the Author

Reviewer #1: The authors responded to all my comments. The main comment related to the diference between the elephant fingers force and comparison with the human hand was addresed rigorously by improving the state of the art. I have no further comments.

Reviewer #2: Thank you for the corrections and well done. I am happy that all of my comments have been adhered to and the manuscript looks ready for publishing.

7. PLOS authors have the option to publish the peer review history of their article (what does this mean?). If published, this will include your full peer review and any attached files.

Reviewer #1: No

Reviewer #2: No

---

## [Editor Report · Acceptance letter]

30 Apr 2024

PONE-D-23-31488R1 

PLOS ONE

Dear Dr. Costes, 

I'm pleased to inform you that your manuscript has been deemed suitable for publication in PLOS ONE. Congratulations! Your manuscript is now being handed over to our production team.

Kind regards, 

on behalf of

Dr. Monika Błaszczyszyn 

Academic Editor

PLOS ONE